# Prior Exposure to Dry-Cured Meat Promotes Resistance to Simulated Gastric Fluid in *Salmonella* Typhimurium

**DOI:** 10.3390/foods8120603

**Published:** 2019-11-21

**Authors:** Yhan S. Mutz, Denes K. A. Rosario, Vinicius S. Castro, Patricia C. Bernardes, Vania M. F. Paschoalin, Carlos A. Conte-Junior

**Affiliations:** 1Institute of Chemistry, Federal University of Rio de Janeiro, Av. Athos da Silveira Ramos, 149, Cidade Universitaria, Rio de Janeiro 21941-909, RJ, Brazil; viniciuscastro@ufrj.br (V.S.C.); paschv@iq.ufrj.br (V.M.F.P.); 2Analytical and Molecular Laboratory Center, Faculty of Veterinary Medicine, Fluminense Federal University, Vital Brazil Filho, 64, Niteroi 24230-340, RJ, Brazil; 3Center for Food Analysis, Technological Development Support Laboratory (LADETEC), Avenida Horácio Macedo, 1281, Polo de Quimica, bloco C, Ilha do Fundão 21941-598, Brazil; 4Department of Food Engineer, Federal University of Espirito Santo, Alto Universitário, s/n, Alegre 29500-000, ES, Brazil; paticbernardes@gmail.com; 5National Institute of Health Quality Control, Oswaldo Cruz Foundation, Rio de Janeiro 21040-900, RJ, Brazil

**Keywords:** acid tolerance response, response surface methodology, brazilian dry-cured loin, rte meat product, hurdle technology, consumer safety

## Abstract

This study assessed if exposure of foodborne *Salmonella*
*enterica* in Brazilian dry-cured loin (BDL) affects pathogen inactivation in simulated gastric fluid (SGF). The acid tolerance responses of three *Salmonella*
*enterica* serovars, Typhimurium, Derby and Panama, were assessed by an acid challenge trial at pH 3.0 for 4 h following pre-adaptation to three conditions: neutral pH, acidic pH (4.5) or BDL matrix. The influence of *Salmonella* exposure temperature and time in the BDL on pathogen gastric fluid resistance was evaluated by the response surface methodology. The *Salmonella* serovars acquired acid tolerance when exposed to the BDL matrix and their response to acid stress was strain-dependent, with *S*. Typhimurium being the most tolerant strain. *S*. Typhimuirum exposed to temperatures >25 °C in the BDL matrix displayed increased resistance to SGF. By using the response surface methodology, it was determined that *S*. Typhimurium becomes less resistant against SGF if maintained in the BDL matrix at temperatures <7 °C, reinforcing the recommendation to store dry-cured meat under refrigeration in order to minimize consumer risks. The results presented herein point to a novel aspect of hurdle technology that should be taken into account to further understand the risks associated with hurdle-stable meat product, such as dry-cured meats, concerning foodborne pathogen contamination.

## 1. Introduction

Drying and curing are part of traditional practices employed to preserve meat and its products. Currently, dry-cured meats have acquired a new interest, as they present distinct and attractive sensorial characteristics [1]. Due to the regional features conferred by manufacturing technologies, dry-cured meat products have become unique, and include products such as Italian Parma ham, Spanish Serrano ham [2] and Brazilian dry-cured loin (BDL), known as “Socol” [3]. Nonetheless, the development of sensory attributes and the shelf life of these products are strongly influenced by their manufacturing process [4,5]. Due to the manufacturing steps of dry-cured meats, such as salting, fermentation, drying and smoking, these meat products offer a harsh environment for pathogens, making them generally regarded as shelf-stable and safe [6]. However, *Salmonella* outbreaks associated with the consumption of these products have been reported [7,8,9,10].

*Salmonella* was the second most common cause of human zoonotic diseases in 2016 in both the United States and the European Union [11,12]. The natural *Salmonella* habitat is the intestinal tract of warm blooded animals, which is the reason why the primary source of contamination for pork and other meats occurs during the slaughter, where cross-contamination between the animal gut and carcass may occur [13]. Contamination of raw material associated with inadequate manufacturing conditions, such as temperature abuses or insufficient maturation, lead to the presence of pathogenic microorganisms in the final meat product [14]. Moreover, the risk of cross-contamination during post-processing steps, such as slicing, is a public health concern, since these products are marketed as ready-to-eat [15].

The foodborne microbiome from dry-cured meat matrices is exposed to several sublethal stresses inherent to the physicochemical characteristics of these meat products, including, but not limited to, moderate acid pH, low water activity (a_w_) and endogenous microbiota growth competition. The combination of these sublethal stresses, acting together as hurdles to pathogen growth, is the premise behind the safety of dry-cured meats (also called hurdle-stable foods), that do not usually undergo thermal or other treatments prior to consumption to eliminate pathogens [16].

On the other hand, *Salmonella* exposed to mildly acid environments can trigger the acid tolerance response, allowing this pathogen to survive lethal acid environments by modulating its protein expression and activity [17]. Furthermore, foodborne pathogen exposure to multiple and/or subsequent stresses raises concerns regarding stress cross-protection, where pre-exposure to a particular stress leads to increased tolerance against a subsequent exposure to a different stress [18]. Such acquired stress tolerance increases food safety concerns regarding *Salmonella* contamination when occurring in dry-cured meat matrices, as this leads to pathogen persistence even at the end of storage periods [19]. Moreover, foodborne pathogen acid adaptation may increase pathogen survival through the human gastrointestinal passage during the host’s digestive process. Pathogen survival in the acid gastric environment is critical for public health, as the acidic stomach environment is considered the first protective host barrier against contaminated food [20]. It is known that pathogens displaying high acid tolerance may show low oral infective doses (cell densities required to cause illness) [21].

Although studies have focused on the survival of *Salmonella* strains throughout the manufacturing and storage of dry-cured meat products [14,19,22], to the best of our knowledge, no studies have reported the survival of *Salmonella* exposed to a contaminated dry-cured meat product in simulated gastric fluid (SGF). Therefore, the present study aimed to evaluate (i) the induction of acid tolerance response of three distinct *Salmonella* strains, (ii) *S.* Typhimurium survival in the BDL matrix, and (iii) *S*. Typhimurium potential acid tolerance triggered by long-term exposure to the BDL matrix under different temperature and time period conditions and its effects on subsequent SGF survival.

## 2. Materials and Methods

### 2.1. Bacterial Strains and Growth Conditions

*Salmonella enterica* serovar Typhimurium (ATCC 14028), *Salmonella enterica* serovar Derby (IOC 4010), and *Salmonella enterica* serovar Panama (IOC 3694) were provided by the Oswaldo Cruz Institute (IOC), belonging to the Oswaldo Cruz Foundation (FIOCRUZ). Cultures were stored in tryptic soy broth (TSB) (BD^®^, NJ, USA) with 20% (*v*/*v*) glycerol at −80 ± 1 °C, while working cultures were maintained at 4 °C and renewed weekly using Hektoen enteric agar (HE) (Liofilchem^®^, Teramo, ITA).

Cell cultivation was performed by transferring a characteristic colony from HE to 10 mL of glucose free-TSB (TSB-G) followed by 24 h incubation at 37 ± 1 °C until reaching the late stationary phase (cell density of 10^8^–10^9^ CFU/mL), confirmed by plating counts on the HE agar. Plating was performed using an Eddy Jet 2 Spiral Plater (IUL Instruments, Barcelona, ESP) and enumeration was carried out using a Flash & go electronic counter (IUL instruments).

### 2.2. Contamination of Brazilian Dry-Cured Loin (BDL) Samples

BDL samples were purchased from a producer at the city of Venda Nova do Imigrante, ES, Brazil (20°19’31.9”S 41°07’56.6”W). Pork loin curing was carried out using available commercial salt (NaCl), 25 g per kg of pork, and spices (mostly black pepper and garlic powder). Ripening was conducted at ambient temperature (17 ± 5 °C, RH 82% ± 7%) for 90 days between September–November 2018. Vacuum-packed dry-cured loin samples were transported at ambient temperature to the laboratory and were cut using a 178 MC/MC-X 3.0 meat slicer (Arbel^®^, SP, BRA) into approximately 1.0 mm thick slices with a superficial area of 23.5 ± 0.9 cm^2^. Sliced BDL samples displayed pH = 5.49 ± 0.11 and a_w_ = 0.86 ± 0.02.

Cells from each *Salmonella* strains were collected by centrifugation at 5580× *g* for 10 min using a Sorvall ST 16 centrifuge (Thermo Fisher, GER) and suspended in 1 mL of saline peptone containing 0.1% casein peptone (Sigma-Aldrich^®^, Darmstadt, GER) and 0.85% NaCl (Sigma-Aldrich^®^, Darmstadt, Germany). Sliced BDL samples were set as individual 50 g portions in sterile polyethylene bags and were inoculated separately with each *Salmonella* strain. BDL slices were individually surface-inoculated with a 500 μL aliquot containing 1.5 × 10^9^ CFU/g of *Salmonella*, confirmed by HE agar plating. The aliquot was spread with a sterile bending glass rod, the slices were air-dried in a laminar flow and the bags sealed.

### 2.3. Acid Tolerance Response Assessment

#### 2.3.1. Strain Pre-Adaptation

Innate and inducible *Salmonella* acid tolerance responses (ATR) were evaluated by adapting cells to the acidified media or the BDL matrix before the acid challenge trial. *Salmonella* strains grown overnight in TSB–G were used as (negative control) non-adapted (NA) cells. Acid-adapted (AA) *Salmonella* cells (positive control) were grown in TSB broth, acidified with HCl 6N (Sigma-Aldrich^®^, Darmstadt, Germany) to reach pH 4.5 until achieving a cell density of 10^8^–10^9^ CFU/mL.

*Salmonella* cells exposed to the BDL matrix (meat stressed *Salmonella* cells or MS) were obtained by recovery from the contaminated BDL slices (Section 2.2). Briefly, contaminated BDL samples were incubated at 35 ± 1 °C overnight and 10 g of each sample were homogenized using a digital stomacher (YK Tecnologia, RS, BRA) with 90 mL of saline peptone. The homogenates were aseptically collected and centrifuged at 5580× *g* for 10 min and the MS pelleted cells were suspended in saline peptone solution. Decimal serial dilutions were plated on HE agar to estimate the number of surviving cells after 24 h. The absence of *Salmonella* in BDL samples prior to exogenous contamination was confirmed by HE agar plating.

#### 2.3.2. Acid Challenge Trial

Each *Salmonella* strain, pre-adapted or not to sublethal conditions (NA, AA or MS), was pelleted by centrifugation at 5580× *g* for 10 min and suspended in saline peptone at neutral pH. The acid challenge trial was performed by transferring 1 mL of each cell suspension to 9 mL of a saline peptone solution adjusted to pH 3.0 ± 0.03 with HCl 6N and incubated at 37 ± 1 °C for 4 h. Appropriate decimal serial dilutions of cell suspension were plated on HE agar in order to count and estimate *Salmonella* inactivation.

### 2.4. Salmonella Inactivation Modeling in BDL and in Simulated Gastric Fluid (SGF)

#### 2.4.1. Experimental Design

The effects of the temperature and time period of *S*. Typhimurium exposure to BDL matrix on (i) bacteria survival in BDL and (ii) bacteria ability to survive subsequent SGF exposure, were evaluated by a central composite rotatable design (CCRD). The experimental layout of the 2^2^ CCRD, with three replicates in the central point, four axial and four factorial points, is displayed in Table 1. The range of time and temperature variables were set to simulate a situation of post-process contamination followed by BDL transportation and consumption. Therefore, the time was set up to 48 h and the temperature aimed to cover from the recommendable indications of storage (cold storage) up to temperature abuses that can occur in transport and storage at ambient temperatures.

#### 2.4.2. *Salmonella* Survival in the BDL Matrix and Resistance to SGF

To perform the 11 experiments designed by the CCRD, *S.* Typhimurium cells recovered from BDL samples—MS cells (Section 2.2 and Section 2.3.1)—at each incubation condition of the CCRD runs were estimated by enumerating surviving cells (Table 1). *S*. Typhimurium inactivation was calculated as decimal logarithm reductions between the initial inoculum and the recovered cells—Log N_0_/N.

The same 11 experimental conditions were used to assess the effect of each exposure condition in the BDL matrix on subsequent SGF *S*. Typhimurium survival (Table 1). After each experiment, 10 g of the BDL samples were homogenized using the stomacher with 90 mL of SGF and incubated at 37 ± 1 °C for 1 h. The SGF comprised 8.3 g proteose peptone (Sigma-Aldrich, Darmstadt, Germany); 2.05 g sodium chloride; 0.6 g potassium phosphate (Sigma-Aldrich, Darmstadt, Germany) 3.5 g D-glucose (Vetec Química Fina Ltd.a, RJ, Brazil), 0.11 g calcium chloride (Vetec Química Fina Ltd.a), 0.37 g potassium chloride (Tedia Company Inc, RJ, Brazil), 0.1 g lysozyme, and 13.3 mg pepsin (Thermo Fisher Scientific, MA, USA) per liter of distilled water [23,24,25]. *S*. Typhimurium enumerations were performed by plating an aliquot of the BDL-SGF cell suspension prior and after the 1 h incubation at 37 °C. *S*. Typhimurium inactivation was calculated by the logarithm difference from initial and final counts Log N_0_/N.

#### 2.4.3. Model Fitting and Performance Evaluation

*S*. Typhimurium inactivation caused by exposure to the BDL matrix or by sequential exposure to the BDL matrix and SGF was evaluated by the response surface methodology (RSM). A second order polynomial model was fitted for each challenge, maintaining only significant (*p* < 0.05) terms. Goodness-of-fit measures were established by the adjusted coefficient of determination (R^2^_adj_), lack-of-fit test (LOF), and mean squared error (MSE). The Shapiro–Wilk test was applied to verify if residuals were normally distributed.

The model performance was assessed through the accuracy factor (A_f_) and bias factor (B_f_) [26], Equations (1) and (2), respectively. To do this, six additional experiments within the tested conditions range were carried out. The additional experiments were used to calculate the performance indices and were not used in the model construction.
(1)$$A_{f}\;=\;\exp(\sqrt{\frac{\sum_{k=1}^{m}{\left(\mathrm{Ln}\;f\left(x^{k}\right)-{\mathrm{Ln}\mu }^{k}\right)}^{2}}{m}})$$
(2)$$B_{f}\;=\;\exp(\frac{\sum_{k=1}^{m}\left(\mathrm{Ln}\;f\left(x^{k}\right)-{\mathrm{Ln}\mu }^{k}\right)}{m})$$
where Ln f(x) represents the predicted values from the model fit, Ln μ, the experimental values and m is the number of experiments.

### 2.5. Statistical Data Analysis

All experiments were performed as three independent biological replicates followed by analytical duplicates. *Salmonella* strain inactivation data were evaluated by a factorial ANOVA and differences between means were detected by Fisher’s LSD, both at 5% significance, using Statistica^®^ v.10 software (Statsoft Inc., OK, USA).

## 3. Results and Discussion

### 3.1. Salmonella Acid Tolerance Response

S. *enterica* inactivation subjected to acid challenge was dependent on the strain (*p* < 0.05), the previous adaptation condition (*p* < 0.05) and the interaction between these two factors (*p* < 0.05). The acid-adapted group displayed lower inactivation (*p* < 0.05) for all tested strains when compared to the other condition counterparts, indicating acid tolerance response (ATR) induction, increasing innate cell resistance (Table 2).

Inactivation of MS cells from all strains was higher than in AA, but lower than in NA cells. Therefore, it seems that pre-exposure to BDL is able to induce the ATR in *Salmonella* cells, although BDL seems to be less efficient than the acidified media in this regard (Table 2). Such an effect can be attributed to the slightly acid pH of the BDL matrix, of around 5.49 ± 0.11, in the described range of ATR induction for stationary phase *Salmonella* cells [27,28].

The main reason for pH lowering in dry-cured meats, besides rigor mortis, can be endogenous muscle enzymes or the action of exogenous microorganism enzymes with the formation of metabolites [2,5], which triggers ATR in MS cells in response to the increased organic acid concentration in the meat matrix. The chemical nature of the acid—organic or inorganic—enrolled in the acquired acid adaptation is among the numerous factors that can affect S. *enterica* ATR induction and intensity [21,29]. However, the hypothesis that the acid tolerance acquired by S. *enterica* exposed to BDL may result from cross-protection from other stresses promoted by the meat matrix, such as low a_w_ (0.86 ± 0.02), cannot be excluded [30]. These abiotic stresses can trigger stress responses sharing the same molecular mechanisms involving the alternative sigma factor σS, which modulates gene expression in osmotic, desiccation and acid stresses [31,32]. Thus, it can be hypothesized that the combination of various stresses exerted by the BDL matrix during *Salmonella* exposure may have induced *Salmonella* cross-protection against the acid challenge [33].

*Salmonella enterica* ATR intensity has been shown to be strain-dependent, since genic expression modulation depends on the genetic background of each organism [34,35]. The *S*. Typhimurium and *S*. Derby strains did not significantly differ for any of the screened pre-adaptations conditions, while the *S*. Panama strain was shown to be more susceptible to acid, displaying higher inactivation for either non-adapted and acid-adapted cells when compared to the other strains. The distinct responses of *S*. Panama, lower than those found for Typhimurium and Derby, indicate that the innate and inducible ATR of this strain is lower than the other assessed strains, which may indicate the absence or lower availability of ATR-mediators and effectors (Table 2).

Although the BDL matrix triggered a less efficient ATR response than the acid pre-adaptation condition, the acquired acid tolerance by foodborne pathogens at any level raises concerns regarding the safety consumption of these meat products. Due to their manufacturing process and physicochemical characteristics, BDL can be considered a typical dry-cured meat product and, like other similar products, such as dry-cured hams, loins, and fermented sausages, it presents low a_w_ and a moderately acidic pH [4,36]. Since dry-cured meat products share many similarities, concerning both manufacturing and physicochemical characteristics, the ATR induced by the food matrix on *S*. *enterica* cells demonstrated herein may be a general phenomenon for dry-cured meat products.

### 3.2. Salmonella Typhimurium Inactivation Modelling and Model Performance Evaluation

*S*. Typhimurium inactivation caused by BDL matrix exposure was linearly influenced by exposure temperature and time, as described in Equation (3), while inactivation by a subsequent 1 h-exposure to SGF at pH 1.5 was linearly influenced by exposure temperature, time and a quadratic effect of temperature, as described in Equation (4). Both were modeled by a multiple regression analysis, maintaining significant terms (*p* < 0.05). The performance indices used to evaluate the goodness-of-fit for the models, as well as the normal distribution of residual data and a model ensuring the required premises for the regression analysis, are displayed in Table 3.
Log (N_0_/N) = 0.536471 + 0.021555 × ET + 0.025494 × EP(3)
Log (N_0_/N) = 2.483493 − 0.068229 × ET + 0.001066 × ET^2^ + 0.017614 × EP(4)
where ET = exposure temperature and EP = exposure time period.

The Shapiro–Wilk test was applied to verify data normality [37]. Data comprising *Salmonella* Typhimurium inactivation after exposure to SGF following BDL or only to the BDL matrix itself, as well as the residuals of both models, were normally distributed (*p* > 0.05) (Table 3).

The non-significant LOF (*p* > 0.05) strongly indicated a good fit for both models, as this test is calculated through the error variance independent of model predictions [38]. Furthermore, both models presented low MSE, which accounts for the remaining variability of the model, including the natural variability of the experiment and systematic errors [39]. The adjusted coefficients of determination (R^2^_adj_) for both models were over 0.85, indicating adequate data variability explanation.

The *S*. Typhimurium inactivation model in BDL presented a B_f_ value close to 1, while the inactivation model in SGF was equal to 1, which indicates a perfect agreement between model predictions and observations [35]. However, as under- and over-predictions can cancel each other out regarding B_f_ values, this index does not provide any sense of accuracy of model prediction [26]. Therefore, the accuracy factor (Af) was also calculated. The A_f_ values for both models were within the acceptable range (1.0–1.3) for a model that accounts for two variables [40], indicating a good accuracy of the inactivation values predicted by the models.

Taken together, the LOF, R^2^_adj_, MSE, B_f_, and A_f_ indices evidenced that the model predicts the true mean values for *S*. Typhimurium inactivation under long-term BDL exposure followed by SGF exposure.

### 3.3. Salmonella Typhimurium Inactivation in BDL and SGF

*S*. Typhimurium inactivation in BDL was linearly influenced by exposure temperature and time (Figure 1A, Figure 2A, and Equation (3)). BDL, like other dry-cured meat products, offers a harsh environment for pathogens, due to its intrinsic physicochemical characteristics, such as low a_w_ and acidic environment, as mentioned previously. These sub-lethal matrix characteristics triggered a decline in the assessed *S*. Typhimurium population, by not attending to minimal growth pathogen requirements, namely pH 4.1 and a_w_ 0.94 [41]. This *Salmonella* inactivation has been previously reported for other dry-cured meat matrices [19,42,43]. However, although the gradual inactivation of *S.* Typhimurium in BDL over time is a certainty, temperature was found to dictate the extent of the inactivation, where gradual increases in temperature induce higher inactivation for a given exposure period (Figure 1A and Figure 2A). However, temperature cannot account for a direct lethal effect on *S.* Typhimurium inactivation, since the temperatures tested herein ranged from 6 to 38 °C and were lower than thermal treatments in order to mimic a contamination of a dry-cured meat product. Instead, temperature effects seem to be indirect, due to the modulation of biochemical reactions, as mesophilic microorganism metabolisms maintained under their optimal growth temperature (25–35 °C) are present in a more activated state when compared to lower temperatures [41,44]. Exposure to these mild temperatures leads to greater energy demands in order to maintain cell homeostasis, which, in turn, leads to further inactivation of *Salmonella* cells retained in the hurdle environmental conditions of dry-cured meat matrices [19].

*S.* Typhimurium inactivation in simulated gastric fluid following BDL exposure is described in Figure 1B, Figure 2B and Equation (4). The prior stresses caused by the harsh BDL matrix environment under different exposure conditions were shown to affect *S*. Typhimurium cell survival. *S*. Typhimurium inactivation in SGF was linearly influenced by temperature and time, and had the influence of a quadratic temperature term as well (Equation (4)). Cells exposed to BDL at cooling temperatures (< 7 °C) presented the highest inactivation rates after SGF exposure (Figure 1B and Figure 2B). The association between BDL exposure in low temperatures and higher inactivation in subsequent SGF conditions may be attributed to the *Salmonella* adaptation mechanism to cooling temperatures. In these low temperatures, *Salmonella* cells change the lipid profile of their membranes to increase membrane fluidity, which, in turn, enhances inactivation in acidic environments, corroborating data previously reported for *Salmonella* Enteritidis grown in low temperatures [25].

*S*. Typhimurium inactivation in SGF was reduced with increasing BDL exposure temperatures. The lowest inactivation zone is observed at temperatures above 25 °C, (Figure 1B and Figure 2B) and cell resistance to acid environments may be attributed to stress cross-protection between the matrix stresses at higher temperatures and acidic exposure. The regulatory proteins/systems that protect *Salmonella* cells against sublethal stresses promoted by the inherent physicochemical characteristics of the BDL matrix lead to cross-protection in the acid environment, mimicking the human gastric fluid faced by foodborne pathogens during host digestive processes [33]. The fact that this effect was more pronounced in increasing BDL exposure temperatures may be attributed to the metabolic activity enhancement of *Salmonella* cells required to maintain cell homeostasis in the BDL matrix, as well as the non-change of the membrane lipid profile [25,44].

The effect of exposure time was significant (Equation (4)) and although no significant interaction with temperature was observed, this parameter shaped *S*. Typhimurium inactivation in SGF. As observed in the iso-inactivation plot (Figure 2B), the lowest inactivation zone points to a critical scenario for the consumption of a contaminated ready-to-eat BDL handled in abusive temperatures. Furthermore, lowering the exposure temperatures lead to decreasing in the time required for *S.* Typhimurium inactivation in SGF (Figure 2B). These results corroborate the European Parliament recommendations in the form of regulation 853/2004/EC for the storage of ready-to-eat meat products under refrigeration [45].

The temperature range tested herein aimed to cover the manufacturer’s recommendation for storage and possible abusive temperatures according to potential environmental product exposure conditions. To further facilitate the dataset interpretation, iso-reduction plots (Figure 2) were constructed to present different *S*. Typhimurium inactivation levels in BDL and SGF as a function of exposure temperature and time, where each line was drawn at a 0.5 Log CFU interval, which corresponds to the uncertainty of the microbiological count method [46]. The data indicate that the sum of inactivation effects from the BDL matrix and simulation of the human gastric environment pointed to higher *Salmonella* survival in the dry-cured matrix with subsequent death in SGF during exposure at cold temperatures (<7 °C); or a sharp death in the matrix with increased resistance to SGF under abusive temperatures (>25 °C). Regarding the concept of food preservation by hurdle technology, Leistner [16] stated that refrigeration may be not always beneficial for products that inhibit microbial growth without the need for this type of storage, such as BDL, as microorganisms can survive for longer periods and may, therefore, increase the possibility of foodborne illness. However, our findings pointed to a reassuring factor for maintaining dry-cured meat under refrigeration, therefore corroborating the established concepts of the hurdle technology, adding a new layer of understanding concerning the risk factors associated with foodborne pathogen contamination of stable meat products, such as dry-cured meats [33].

## 4. Conclusions

*Salmonella* contamination during the manufacturing or storage of dry-cured meats may represent an unexpected and serious concern regarding safety risks in the consumption of these ready-to-eat foods. Exposure to the BDL matrix, a Brazilian dry-cured loin, induces acid tolerance in *Salmonella enterica* Typhimurium, Derby and Panama, while also increasing *S*. Typhimurium survival in simulated gastric fluid. The increased survival of *S*. Typhimurium to the simulated gastric environment may be related to the acid tolerance response and the cross-protection mechanisms induced by the physicochemical characteristics of the dry-cured meat matrix on pathogen physiology. In addition, food poisoning risks may be further outlined, since such a phenomenon is highly dependent on the dry-cured product storage conditions, that is, time and temperature, when hosting the pathogens.

The results of the present study corroborate the concept of food preservation by the hurdle technology, by taking into account resistance against the gastric conditions of a foodborne pathogen following different long-term exposure conditions in a hurdle-stable matrix. These observations may have an important impact on public health, as tolerant cells can increase the chance of *S*. Typhimurium overcoming host digestive gastric barriers. Further studies concerning the virulence determinants of *Salmonella* cells stressed by dry-cured meat matrices are suggested.

## Figures and Tables

**Figure 1 foods-08-00603-f001:**
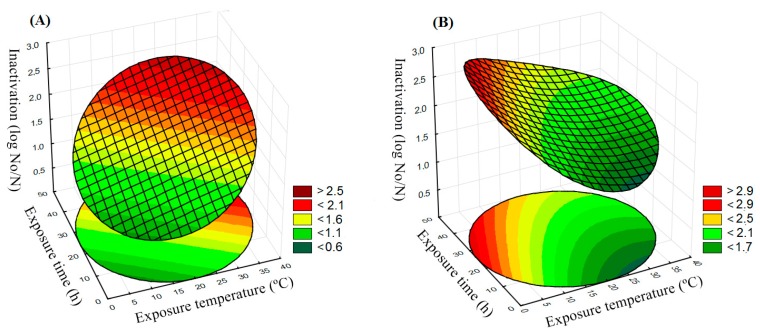
*Salmonella* Typhimurium inactivation response surface (Log (N_0_/N)) in: (**A**) Brazilian dry-cured loin(BDL) under different exposure temperatures and periods in the dry-cured matrix; (**B**) One-hour simulated gastric fluid (SGF)— at pH 1.5 following exposure under different temperatures and times in the BDL matrix.

**Figure 2 foods-08-00603-f002:**
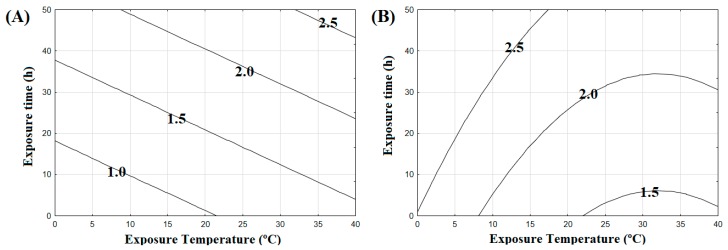
Iso-reduction plot drawn from the developed models, where each line describes an equivalent Log reduction of *S*. Typhimurium in (**A**) Brazilian dry-cured loin (BDL) as a function of temperature and time exposure (Equation (4)); (**B**) one-hour simulated gastric fluid (SGF)—exposure at pH 1.5 following Brazilian dry-cured (BDL) as a function of temperature and time exposure.

**Table 1 foods-08-00603-t001:** Central composite rotatable design (CCRD) arrangement and decimal reduction of *Salmonella* Typhimurium in the Brazilian dry-cured loin (BDL) matrix and after 1 h exposure to simulated gastric fluid.

Run	Exposure Temperature (°C)	Exposure Time (h)	Inactivation in BDL (log N_0_/N)	Inactivation in SGF (log N_0_/N)
1	10.0	7.1	1.08	1.92
2	10.0	41.0	1.63	2.78
3	33.0	7.1	1.36	1.58
4	33.0	41.0	2.09	2.09
5	5.7	24.0	1.21	2.53
6	38.2	24.0	2.08	1.84
7	22.0	0.0	1.03	1.50
8	22.0	48.0	2.57	2.22
9	22.0	24.0	1.57	2.01
10	22.0	24.0	1.54	1.91
11	22.0	24.0	1.65	1.93

**Table 2 foods-08-00603-t002:** *Salmonella enterica* inactivation after the acid challenge trial.

Adaptation	Strain	Decimal Reduction
Non-adapted cells (NA)	Panama	1.55 ± 0.09 ^a^
	Derby	1.40 ± 0.11 ^a,b^
	Typhimurium	1.31 ± 0.08 ^b^
Meat-stressed cells (MS)	Panama	0.96 ± 0.09 ^c^
	Derby	0.77 ± 0.20 ^c,d^
	Typhimurium	0.77 ± 0.26 ^c,d^
Acid-adapted cells (AA)	Panama	0.67 ± 0.09 ^d^
	Derby	0.11 ± 0.02 ^e^
	Typhimurium	0.03 ± 0.02 ^e^

Non-adapted, meat-stressed or acid-adapted cells were shifted to an acidified culture media at pH 3 for 4 h. Decimal reductions are expressed as the means and standard deviation of three biological replicates, performed in duplicate (*n* = 3 × 2). Different subscript letters indicate differences at a significance level of *p* < 0.05 by the Fisher LSD test.

**Table 3 foods-08-00603-t003:** Distribution normality and performance of *S*. Typhimurium inactivation indices in the BDL matrix followed by SGF exposure.

Inactivation Model	Distribution Normality *	Residual Distribution Normality *	*R* ^2^ _adj_	Mse	Lack-of-Fit Test	A_f_	B_f_
Inactivation in BDL	Normal (0.48)	Normal (0.47)	0.87	0.021	0.08	1.12	1.03
Inactivation in SGF	Normal (0.50)	Normal (0.55)	0.93	0.006	0.20	1.10	1.00

* *p* values obtained by the Shapiro–Wilk tests. R^2^_adj_: adjusted coefficient of determination; MSE: mean squared error; A_f_: accuracy factor; B_f_: Bias factor; SGF; simulated gastric fluid.

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
