# Peer review of "Prior Exposure to Dry-Cured Meat Promotes Resistance to Simulated Gastric Fluid in Salmonella Typhimurium"

_foods, 2019, doi:10.3390/foods8120603_

Round 1

Reviewer 1 Report

I think the paper is very clear and the conclusion answers the objectives of the study.

Author Response

The manuscript was submitted to an English revision by a native English professional.

Reviewer 2 Report

Very-well designed, satisfied scientific paper which may be handled for a few minor changes.

Abstract: Should be rearranged. Methodology is unclear in abstract

Line 22: It is better to write “subsp” after “enterica

Line 43: “shelf life” instead of shelf lives

Line 46: It is better to add the abbreviation of “generally regarded as safe” as “GRAS” in brackets

Line 97: “BDL samples were purchased at the city of…” Did you receive the pre-prepared BDL samples or the researchers produced in the experiment design? That point is not clear.

Line 109: why 1.5 x 109 CFU/g of Salmonella?

Author Response

Very-well designed, satisfied scientific paper which may be handled for a few minor changes.

We would like to thank the editor/reviewers for their insights and thoughtful critique of our manuscript.

Q: Abstract: Should be rearranged. Methodology is unclear in abstract

A: A methodology section was included in the abstract (L21-26), as requested.

Q: Line 22: It is better to write “subsp” after “enterica”

A: We have added this, as requested.

Q: Line 43: “shelf life” instead of shelf lives

A: Thank you. The expression was replaced.

Q: Line 46: It is better to add the abbreviation of “generally regarded as safe” as “GRAS” in brackets

A: This term was included, as requested.

Q: Line 97: “BDL samples were purchased at the city of…” Did you receive the pre-prepared BDL samples or the researchers produced in the experiment design? That point is not clear.

A: The BDL was bought ready to use, produced in the city by a local manufacturer. We have altered the text to make this clearer.

Q: Line 109: why 1.5 x 109 CFU/g of Salmonella?

A: The initial bacterial load was aimed to be enough to assess the effects of two consecutive steps, long term exposure to BDL and subsequent exposure to SGF. Thus, a high inoculum was used.

Reviewer 3 Report

The authors studied the acidic resistance of Salmonella in several conditions. They compared the acidic resistance of three strains of Salmonella pre-adapted (in laboratory media or in dry cured meat) or not. They focus on one strain of Salmonella and characterised its resistance in simulated gastric fluid after exposure to dry cured meat. For this last experiment, they studied the effect of several temperatures and times, they studied these factors by carrying experimental design and modelling. The manuscript is well constructed, easy enough to read and presented original results but the following remarks can be made.
In the title and along the text, the word matrix could be suppressed

Line 21: a dry cured meat matrix can be suppressed

Line 37: a new relevance? A new interest

Lines 41-42: Is there really a fermentation step in dry cured meat? Lactic acid bacteria (LAB) are often in low number in these products, no sugar is added, the drop in pH is mainly associated to the activity of endogenous muscle enzymes activity. Could you check this point?

Line 46, the word exogenous can be suppressed

Lines 78-79: please rephrase, as written it seems that SGC is obtained from a contaminated dry cured meat

Lines 86-87: please write the name of all the salmonella for the first citation and then use abbreviation

Line 96, the title will be more comprehensible: Contamination of Brazilian dry-cured loin (BDL) samples

Lines 97 to 100: too much details on the city of the purchase, on time of manufacturing (September…), on transportation... Could you simplify this part? Do you know the quantity of salt added? the spices? You did not comment an osmotic stress due to salt?

Line 117: could you put in bracket (the positive control)

Line 119: could you put in bracket (meat stressed Salmonella cells, MS)

Table 1: I do not understand the experimental design, chooses of temperatures, of times and the presence in this table of factors studied and results. Could you explain?

In table 1 and in all the text, the word permanence is not suitable, could you replace it, exposure?

Line 146: replace every by each

Line 162 was instead of were

Table 2: previous adaptation: previous not useful

Line 200: discussion on LAB, see my remark above on fermentation, what is the LAB population in these kind of products? Dry cured meats are different from dry fermented sausages

Line 240: PP; exposure time? EP in the equation?

Legend Figure 1, replace a and b, by A and B

The curves in Figure 2 are not readable, could you modify?

Line 352: Our results corroborate ….

Line 354: These observations may have an …

The references should be checked, ref 3: supress parenthesis, ref 7: 12:i-?

Author Response

The authors studied the acidic resistance of Salmonella in several conditions. They compared the acidic resistance of three strains of Salmonella pre-adapted (in laboratory media or in dry cured meat) or not. They focus on one strain of Salmonella and characterized its resistance in simulated gastric fluid after exposure to dry cured meat. For this last experiment, they studied the effect of several temperatures and times, they studied these factors by carrying experimental design and modelling. The manuscript is well constructed, easy enough to read and presented original results but the following remarks can be made.

We would like to thank the editor/reviewers for their insights and thoughtful critique of our manuscript.

Q: In the title and along the text, the word matrix could be suppressed

A: Thank you. This word was suppressed, as requested.

Q: Line 21: a dry cured meat matrix can be suppressed

A: Thank you. This expression was suppressed, as requested.

Q: Line 37: a new relevance? A new interest

A: Thank you for the suggestion. This change was made in the revised manuscript.

Q: Lines 41-42: Is there really a fermentation step in dry cured meat? Lactic acid bacteria (LAB) are often in low number in these products, no sugar is added, the drop in pH is mainly associated to the activity of endogenous muscle enzymes activity. Could you check this point?

A: Thank you for bringing this point up. Indeed no published studies assessing BDl microbiota are available, as this product is an artisanal product with unexplored potential. However, due to its similarities with other similar products, concerning raw material and manufacturing process, it was inferred that the LAB population and that of other fermentative microorganisms play a role in product development and physicochemical characteristics, such as pH and aw. However, the sentence was removed to avoid misinterpretation about unpublished findings. Our research group has submitted a paper concerning BDL microbiota, currently under review.

Q: Line 46, the word exogenous can be suppressed

A: This word was suppressed, as requested

Q: Lines 78-79: please rephrase, as written it seems that SGC is obtained from a contaminated dry cured meat

A: This sentence was rephrased to avoid misinterpretations.

Q: Lines 86-87: please write the name of all the salmonella for the first citation and then use abbreviation

A: Thank you for the suggestion, this was done, as requested.

Q: Line 96, the title will be more comprehensible: Contamination of Brazilian dry-cured loin (BDL) samples

A: Thank you for the suggestion, this was done, as requested.

Q: Lines 97 to 100: too much details on the city of the purchase, on time of manufacturing (September…), on transportation... Could you simplify this part? Do you know the quantity of salt added? the spices? You did not comment an osmotic stress due to salt?

A: Thank you for the question. The detailed information on the location and date of production are intended to inform about the temperature and humidity during the ripening process, as this is conducted at ambient temperature. The amount of salt was added to the text. Furthermore, BDL production is an artisanal process and its manufacturing is carried out through dry-curing. Due to the similarities of the raw material and manufacturing process between BDL and other known cured meats, osmotic stress, common stress among this class of products (Mutz et al., 2019), can be inferred

Mutz, Y. D. S., Rosario, D. K. A., Paschoalin, V. M. F., & Conte-Junior, C. A. (2019). Salmonella enterica: A hidden risk for dry-cured meat consumption? Critical reviews in food science and nutrition, 1-15

Q: Line 117: could you put in bracket (the positive control)

A: This has been done, as requested.

Line 119: could you put in bracket (meat stressed Salmonella cells, MS)

A: This has been done, as requested.

Q: Table 1: do not understand the experimental design, chooses of temperatures, of times and the presence in this table of factors studied and results. Could you explain?

A: Thank you for the question. The following sentences were added to the text to explain the choice of the factors (L148-152). ”The range of the time and temperature variables were set to simulate a situation of post-process contamination followed by BDL transportation and consumption. Therefore, the time was set up to 48h and the temperature aimed to cover from the recommendable indications of storage (cold storage) up to temperature abuses that can occur in transport and storage at ambient temperatures.”

Q: In table 1 and in all the text, the word permanence is not suitable, could you replace it, exposure?

A: Thank you.  This word was replaced in the text and table.

Q: Line 146: replace every by each

A: This has been altered, as requested.

Q: Line 162 was instead of were

A: This has been altered, as requested.

Q: Table 2: previous adaptation: previous not useful

A: This has been altered, as requested.

Q: Line 200: discussion on LAB, see my remark above on fermentation, what is the LAB population in these kind of products? Dry cured meats are different from dry fermented sausages

A: Thank you for the observation. A change has been made to acknowledge the drop in pH by proteolytic enzyme activity.

Q: Line 240: PP; exposure time? EP in the equation?

A: Thank you for the observation. This has been modified in the revised manuscript.

Q: Legend Figure 1, replace a and b, by A and B

A: This has been altered, as requested.

Q: The curves in Figure 2 are not readable, could you modify?

A: Figure 2 has been modified for clarity, as requested.

Q: Line 352: Our results corroborate ….

A: This has been altered, as requested.

Q: Line 354: These observations may have an …

A: This has been altered, as requested.

Q: The references should be checked, ref 3: supress parenthesis, ref 7: 12:i-?

A: All references were checked.  Concerning reference 7, 12:i- is the name of the Salmonella variant.